# Healthy behaviours, treatment, and control status of diagnosed hypertension and diabetes among the government nurses and para-health professionals of Bangladesh: A cross-sectional study

**Lingkan Barua** **\*, Palash Chandra Banik, Mithila Faruque**

Department of Noncommunicable Diseases, Bangladesh University of Health Sciences (BUHS), Dhaka, Bangladesh

\* lingkanbarua@gmail.com

## Abstract

Chronic illness among health professionals (HPs) is rarely reported due to idealistic views of their role in treating and fighting diseases. This creates a gap mainly due to a lack of research on them, resulting in insufficient data at the national level, especially in Bangladesh. In this circumstance, we analyzed the data of senior staff nurses (SSNs) and para-health professionals (PHPs) to assess their healthy bahaviours, treatment, and control status of hypertension (HTN) and diabetes mellitus (DM). It was a cross-sectional study that used the census as a sampling technique. The study site was a medical university in Bangladesh located in the capital city of Dhaka. A total of 1942 government-employed health professionals working at Upazila Health Complexes participated and completed both the questionnaire and physical measurements with a response rate of 100%. Among them, 1912 (SSNs = 938 and PHPs = 974) remained for analysis after data cleaning. The prevalence of self-screening (HTN, 97.4%; DM, 81.5%), diagnosis (HTN, 20.5%; DM,15.3%), treatment (HTN, 88.7%; DM, 83.7%) and control status (HTN, 63.7%; DM, 31%) did not reveal any notable differences between SSNs and PHPs. Most of the HPs with HTN and DM failed to maintain adequate physical activity (87.4%; 86.2%), fruit and/or vegetable intake (60.7%; 59%), and healthy body weight (60.5%; 54%) respectively. Only avoidance of smoking showed a significant association with the professional categories in both hypertensives (AOR, 7.98; p = 0.001) and diabetics (AOR, 14.78; p<0.001). Although working in the field of primary health care and involved in assisting patient management, control of HTN, DM and their risk factors is not satisfactory among the SSNs and PHPs of Bangladesh. Future interventions should focus on smoking, diet, and physical activity to reduce HTN and DM in the HPs of Bangladesh.

**Data Availability Statement:** The data is available at Zenodo: https://doi.org/10.5281/zenodo.7421378.

**Funding:** The authors received no specific funding for this work.

**Competing interests:** The authors have declared that no competing interests exist.

## Introduction

Noncommunicable diseases (NCDs) are the leading cause of death and disability in developing countries [1]. An important aspect of NCD care is the 'continuity of care' that starts with screening followed by diagnosis, treatment and continues as a follow-up. However, the health systems of many developing countries underestimate the severity and prevalence of chronic diseases [2]. As a result, many chronic diseases remain without diagnosis which poses a risk of serious complications and increases the rate of mortality. In the general population, health beliefs, accessibility, and affordability act as a factor that impacts their health-seeking behavior, including screening [3]. In the case of health professionals, the scenario may be different as a previous study found a high prevalence of hypertension (HTN), diabetes mellitus (DM) and other NCD risk factors among the health professionals of Bangladesh [4]. However, their healthy behavioral practice has not yet been evaluated in any context that needs to be explored.

Health professionals (HPs) are at high risk for developing chronic diseases due to their mode of service delivery. They are exposed to unavoidable physical and mental stress resulting from shifting duties, overtime, providing medical services under life-and-death circumstances, etc. [5]. In countries like Bangladesh, most of them deliver health services in low-resource settings with disproportionate distribution of skilled workforce. So, they adapt themselves physically and mentally in an overburdened, understaffed, and insufficiently equipped healthcare setting that puts unusual stress [6]. However, their health issues are rarely addressed by the superior authorities and policymakers. This gap is mainly due to the lack of research on them, resulting in insufficient data at the national level. In Bangladesh, cardiovascular disease (CVD) is the predominant and downstream cause of mortality from chronic diseases [7]. HTN and DM are the two main intermediate risk factors of CVD which have the higher burden and are relatively easy to screen in a low-resource setting. Although easy to detect and prevent, previous research found a notable prevalence of these two risk factors among health professionals in Bangladesh [4,8]. Global research also reported a higher prevalence of HTN and DM among HPs [9–13]. Therefore, this study aimed to assess healthy behaviours, treatment, and control status among government-employed health professionals who were diagnosed with DM and HTN.

## Materials and methods

The detailed study methodology was previously published elsewhere [4] and here we have described briefly.

### Study design, setting and population

This cross-sectional study was conducted in a medical university among 1942 government-employed senior staff nurses (SSNs) and para-health professionals (PHPs) who were sent by the Non-communicable Disease Control cell of the Directorate General of Health Services to participate in a 3-day training on NCD management during November 2017 to May 2018. Here PHPs included sub-assistant community medical officers, medical technologists, pharmacists and sanitary inspectors. This study was a census as all of these trainees participated in the study.

### Data collection procedure

To collect data, we used a pretested self-administered questionnaire that was a modified STEP-wise approach to Surveillance (STEPS) of the NCD risk factors questionnaire of the World Health Organization (WHO, Version 3.2) [14]. To reduce recall bias, enough time

was given to respondents and pictorial 'show cards' were used to make them understand the risk factors. In the first step, we collected information about the participants' sociodemographic, behavioral, and metabolic risk factors. In the second step, anthropometry (height, weight, waist circumference, and hip circumference) and blood pressure (BP) were measured. To measure height, participants were requested to remove their footwear and headgear, if any. After that they were asked to stand with feet together, heels against the floor, knees straight, eyes at the same level as the ear, and looking straight ahead facing the interviewer. The height was measured by using a height- measuring plastic tape in centimeters (cm). To measure weight, participants had to stand still on a weighing scale putting on a firm and flat surface following removal of their footwear and light casual clothing. Then weight was measured by the portable weighing scale in kilogram (kg) (CAMRY, model BR2016, Hong Kong). Waist circumference was measured by a plastic tape placed horizontally midway between the two points of the lowest rib margin and the top of the iliac crest [15]. The BP was measured two times using an aneroid sphygmomanometer, 1st measurement was taken after 15 minutes of rest time, and the 2nd measurement was taken three minutes after the 1st measurement. The mean of the two measurements was used to determine the final BP. All the physical measurements were carried out by the investigators themselves maintaining adequate privacy. The third step was to measure their capillary blood glucose (after a minimum of eight hours of fasting) using a standard glucometer (GlucoLeaderTM Enhance, HMD BioMedical, Taiwan) with aseptic precautions. The first (STEP 1) and second step (STEP 2) was completed on 1st day of training, and 3rd step (STEP 3) was completed on 2nd day of training (Fig 1). The questionnaire was distributed on the first day of the training.

## Ascertainment of the key variables

The current tobacco use, current alcohol consumption, inadequate fruit and/vegetable intake, inadequate physical activity, and healthy body weight were categorized as per the definition used in the 'Noncommunicable disease risk factors survey Bangladesh 2010' [15]. Intake of added salt was defined as taking dietary salt during eating a meal. Hypertension was defined as systolic BP $\geq$ 140 mmHg and/or diastolic BP $\geq$ 90 mmHg and/or on antihypertensive treatment for elevated blood pressure [16]. Diabetes was defined as fasting blood glucose is $\geq$ 7.0 mmol/L and/or on antidiabetic treatment for raised blood glucose [17]. Here, controlled HTN was defined as systolic BP < 140 mmHg and/or diastolic BP < 90 mmHg in an HP with diagnosed HTN. Again, controlled DM was defined as fasting capillary glucose of <7 mmol/l in an HP with diagnosed DM.

## Ethical consideration

Data were collected after informed written consent was obtained. The study was permitted by the Ethical Review Committee of Bangladesh University of Health Sciences (Memo number: BUHS/BIO/EA/17/2) and carried out according to the Declaration of Helsinki (October 2013).

## Statistical analysis

The final analysis consisted of 1912 responses out of 1942 HPs due to incompleteness or inconsistency. Data were analyzed using Statistical Product and Service Solutions (version 20.0) for Windows (SPSS, Inc. Chicago, IL.USA). Descriptive statistics were presented as the frequency, percentage, and median with interquartile range as appropriate. The impact of categories of health professions (SSNs, PHPs) on self-screening, diagnosis, healthy

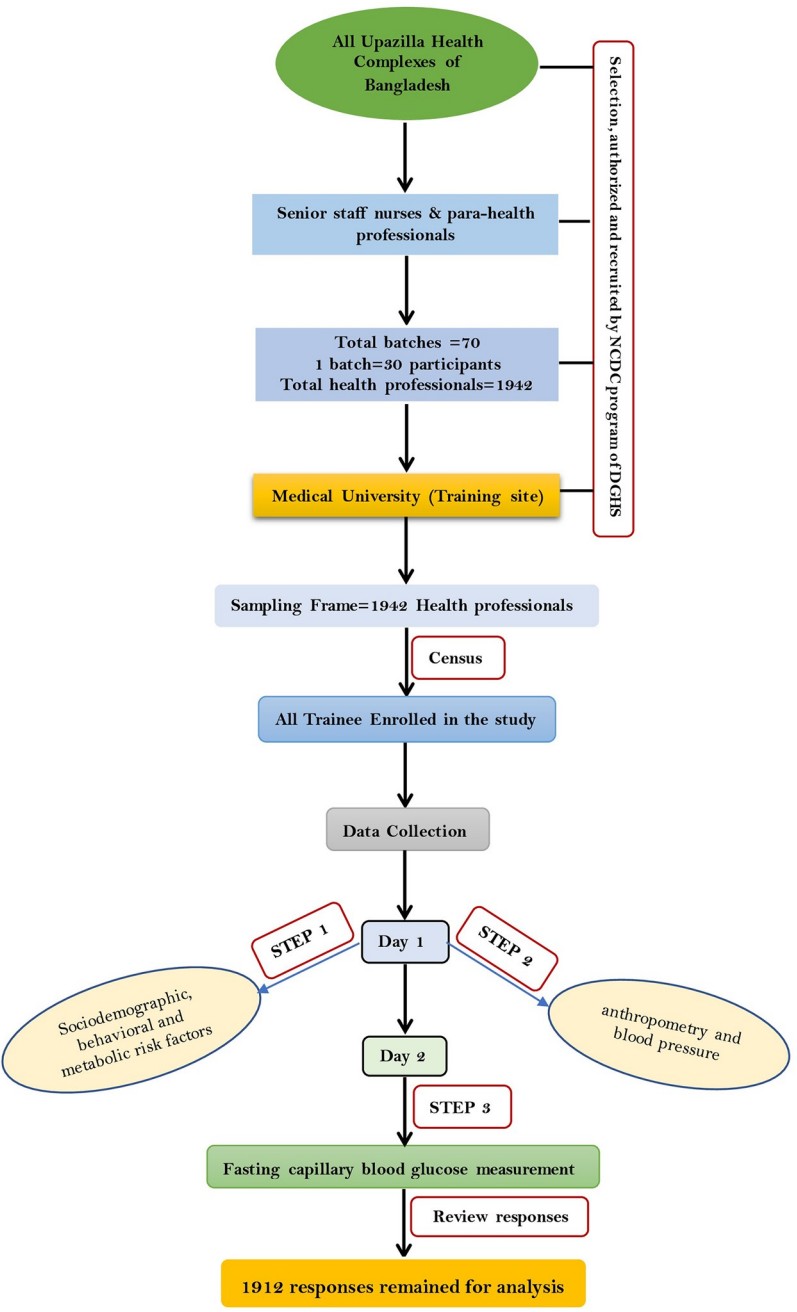

**Fig 1. Sample recruitment, sampling, and data collection technique.**

behaviours, treatment, and control of HTN, as well as DM was evaluated using the Chi-square test and subsequent binary logistic regression. The potential variables to include in the logistic regression were screened based on p≤0.25 yielded as an output of the Chi-square test. We adjusted all variables included in logistic regression to reduce the effect of residual confounders. All precision estimates were presented at a 95% confidence interval. The findings were considered statistically significant at the level of p<0.05 (2-sided).

## Results

A total of 1942 HPs completed both the questionnaire and physical measurements with a response rate of 100%. Among them, 1912 remained for analysis after data cleaning (30 responses were incomplete or inconsistent). A nearly equal proportion of SSNs (n = 938) and PHPs (n = 974) were included in this study who were from all Upazila Health Complexes in the country covering 8 divisions and 64 districts. Their highest educational qualification was a diploma and nearly half of them were comparatively young adults with a 13.8±9.5 years of mean duration in government service [4].

The prevalence of self-screening (HTN, 97.4%; DM, 81.5%), diagnosis (HTN, 20.5%; DM,15.3%), treatment (HTN, 88.7%; DM, 83.7%) and control (HTN, 63.7%; DM, 31%) status did not identify statistically significant differences between SSNs and PHPs. However, treatment and control domains of HTN and DM showed opposite findings between the professional groups. The SSNs showed a higher proportion for those who received treatment (91.4% vs. 86.2%) and controlled (67.8% vs. 58.9%) their BP than PHPs. On the other hand, PHPs were comparatively higher in proportion than SSNs in terms of treatment and control of DM (Tables 1 and 2).

Regarding healthy behaviours among those diagnosed with HTN and DM, most of the HPs failed to maintain adequate physical activity (87.4%; 86.2%), fruit and/vegetable intake (60.7%;

**Table 1. Healthy behaviours, treatment, and controlled status of diagnosed HTN among the SSNs and PHPs of Bangladesh (N = 1912).**

| Variables | Overall (n = 1912) | | | SSNs (n = 938) | | | PHPs (n = 974) | | | P of $\chi^2$-test | AOR | P of logistic regression |
|---|---|---|---|---|---|---|---|---|---|---|---|---|
| | n | % | 95% CI | n | % | 95% CI | n | % | 95% CI | | | |
| Screened for HTN at least once in lifetime | 1863 | 97.4 | 96.7–98.1 | 917 | 97.8 | 96.9–98.7 | 946 | 97.1 | 96.0–98.2 | 0.38 | - | - |
| Days since last measurement of BP* | 30 (15–60) | | | 30 (15–60) | | | 30 (15–60) | | | | - | - |
| HTN diagnosed while measured | 382 | 20.5 | 18.7–22.3 | 187 | 20.4 | 17.8–23.0 | 195 | 20.6 | 18.0–23.2 | 0.91 | - | - |
| Healthy behaviours | Overall (n = 382) | | | SSNs (n = 187) | | | PHPs (n = 195) | | | P of $\chi^2$-test | AOR | P of logistic regression |
| | n | % | 95% CI | n | % | 95% CI | n | % | 95% CI | | | |
| Avoid current smoking (within last 30 days) ** | 355 | 92.9 | 90.3–95.5 | 184 | 98.4 | 96.6–100.2 | 171 | 87.7 | 83.1–92.3 | <0.001 | 7.98 | 0.001 |
| Avoid current smokeless tobacco (within last 30 days) | 269 | 96.6 | 94.8–98.4 | 181 | 96.8 | 94.3–99.3 | 188 | 96.4 | 93.8–99.0 | 0.84 | - | - |
| Avoid current alcohol intake (within last 30 days) | 378 | 99.0 | 98.0–100.0 | 186 | 99.5 | 98.5–100.5 | 192 | 98.5 | 96.8–100.2 | 0.62 | - | - |
| Adequate fruit and/or vegetables intake† | 150 | 39.3 | 34.4–44.2 | 75 | 40.1 | 33.1–47.1 | 75 | 38.5 | 31.7–45.3 | 0.74 | - | - |
| Avoid added salt intake** | 271 | 70.9 | 66.3–75.5 | 124 | 66.3 | 59.5–73.1 | 147 | 75.4 | 69.4–81.4 | 0.05 | 0.63 | 0.05 |
| Adequate physical activity‡** | 48 | 12.6 | 9.3–15.9 | 19 | 10.2 | 5.9–14.5 | 29 | 14.9 | 9.9–19.9 | 0.17 | 1.73 | 0.09 |
| Healthy body weight maintained§ | 151 | 39.5 | 34.6–44.4 | 70 | 37.4 | 30.5–44.3 | 81 | 41.5 | 34.6–48.4 | 0.41 | - | - |
| Treated for HTN those diagnosed** | 339 | 88.7 | 85.5–91.9 | 171 | 91.4 | 87.4–95.4 | 168 | 86.2 | 81.4–91.0 | 0.10 | 0.62 | 0.16 |
| Blood pressure controlled among those treated** | 215 | 63.4 | 58.3–68.5 | 116 | 67.8 | 60.8–74.8 | 99 | 58.9 | 51.5–66.3 | 0.09 | 0.67 | 0.09 |

AOR, adjusted odds ratio; BP, blood pressure; HTN, hypertension; PHPs, para-health professionals; SSNs, senior stuff nurses.

*Presented as median and interquartile range.

†WHO recommendation of fruit/vegetables intake ≥ 5 servings per day.

‡WHO recommendation of Physical activity > 600 MET-minutes/week.

§Body mass index < 25 kg/m2.

Systolic blood pressure < 140 mmHg and/or diastolic blood pressure < 90 mmHg; n = 339.

¶P-value significant at a threshold of <0.05 yielded using Chi-square test & binary logistic regression analysis.

**Included in the binary logistic regression based on P-value ≤0.25 yielded using Chi-square test.

**Table 2. Healthy behaviours, treatment and controlled status of diagnosed DM among the SSNs and PHPs of Bangladesh (N = 1912).**

| Variables | Overall (n = 1912) | | | SSNs (n = 938) | | | PHPs (n = 974) | | | *P* of χ²-test | AOR | *P* of logistic regression |
|---|---|---|---|---|---|---|---|---|---|---|---|---|
| | n | % | 95% CI | n | % | 95% CI | n | % | 95% CI | | | |
| Screened for DM at least once in lifetime | 1559 | 81.5 | 79.8–83.2 | 757 | 80.7 | 78.2–83.2 | 802 | 82.3 | 79.9–84.7 | 0.36 | - | - |
| Days since last measurement of BG* | | 60 (30–60) | | | 30 (15–60) | | | 30 (15–60) | | | | |
| DM diagnosed while measured | 239 | 15.3 | 13.5–17.1 | 128 | 16.9 | 14.2–19.6.5 | 111 | 13.8 | 11.4–16.2 | 0.91 | - | - |
| Healthy behaviours | Overall (n = 239) | | | SSNs (n = 128) | | | PHPs (n = 111) | | | *P* of χ2-test | AOR | *P* of logistic regression |
| | n | % | 95% CI | n | % | 95% CI | n | % | 95% CI | | | |
| Avoid current smoking (within last 30 days) ** | 223 | 93.3 | 90.1–96.5 | 126 | 98.4 | 96.2–100.6 | 97 | 87.4 | 81.2–93.6 | 0.001 | 14.78 | <0.001 |
| Avoid current smokeless tobacco (within last 30 days) | 229 | 95.8 | 93.3–98.3 | 124 | 96.9 | 93.9–99.9 | 105 | 94.6 | 90.4–98.8 | 0.52 | - | - |
| Avoid current alcohol intake (within last 30 days) | 237 | 99.2 | 98.1–100.3 | 127 | 99.2 | 97.7–100.7 | 110 | 99.1 | 97.3–100.9 | 1.00 | - | - |
| Adequate fruit and/or vegetables intake†** | 98 | 41.0 | 34.8–47.2 | 46 | 35.9 | 27.6–44.2 | 52 | 46.8 | 37.5–56.1 | 0.09 | 1.07 | 0.508 |
| Avoid added salt intake** | 171 | 71.5 | 65.8–77.2 | 85 | 66.4 | 58.2–74.1 | 86 | 77.5 | 69.7–85.3 | 0.06 | 0.65 | <0.001 |
| Adequate physical activity‡** | 33 | 13.8 | 9.4–18.2 | 13 | 10.2 | 5.0–15.4 | 20 | 18.0 | 10.9–25.1 | 0.08 | 2.44 | <0.001 |
| Healthy body weight maintained§** | 110 | 46.0 | 39.7–52.3 | 52 | 40.6 | 32.1–49.1 | 58 | 52.3 | 34.6–48.4 | 0.07 | 1.25 | 0.021 |
| Treated for DM those diagnosed | 200 | 83.7 | 79.0–88.4 | 106 | 82.8 | 76.2–89.3 | 94 | 84.7 | 78.0–91.4 | 0.67 | - | - |
| DM controlled among those treated | 62 | 31.0 | 24.6–37.4 | 30 | 28.3 | 19.7–36.9 | 32 | 34.0 | 24.4–43.6 | 0.38 | - | - |

AOR, adjusted odds ratio; BG, blood glucose; DM, diabetes mellitus; PHPs, para-health professionals; SSNs, senior stuff nurses.

*Presented as median and interquartile range.

†WHO recommendation of fruit/vegetables intake > 5 servings per day.

‡Physical activity > 600 MET-minutes/week.

§Body mass index < 25 kg/m2.

Fasting plasma glucose of <7 mmol/l; n = 200.

¶P-value significant at a threshold of <0.05 yielded using Chi-square test & binary logistic regression.

**Included in the binary logistic regression based on P-value ≤0.25 yielded using Chi-square test.

59%) and healthy body weight (60.5%; 54%), respectively. A highly significant association was found between professional categories and avoidance of smoking in hypertensive patients (AOR, 7.98; p = 0.001) and diabetics (AOR, 14.78; p<0.001) (Tables 1 and 2).

## Discussion

In summary, the current study found that most HPs screened themselves and received treatment following the diagnosis of HTN and DM. However, their control rate was poor for both diseases, especially for DM. This failure to control HTN and DM is possible due to unhealthy lifestyle choices elucidated in this study.

Our study found that self-screening for HTN and DM is very common among HPs (97.4% for HTN, 81.5% for DM) supporting the previous finding [18]. Again, the proportion of HTN screening was higher than that DM screening which is in line with previous studies [18,19]. Although both pieces of evidences [18,19] aligned with our findings, they studied among HPs rather than primary care. In general, despite health facilities, this is not surprising that most HPs were screened themselves at least once in their life because they work in a health care setting where scopes are abundant and most of them had a diploma in their respective field of work justifying their knowledge about the fact [20]. However, comparatively high cost,

invasive technique, and time-consuming gold standard method make HPs reluctant to screen DM than the HTN.

We found job-specific differences in the domain of treatment and control which was statistically non-significant. Here, the rate for treatment and control of HTN was higher in SSN whereas DM treatment and control were higher among PHPs. The exact cause of such differences among HPs based on their profession category is not known. Although they are working in the same healthcare setting, possibly their knowledge, attitude and practice do not converge [4]. This is because the educational qualification and clinical training of SSN (Diploma/BSc/MSc) are completely different from the PHPs (Diploma) and their clinical role is also varied in Bangladesh. Here, SSNs were mostly involved in indoor patient care, and Sub-assistant community medical officers (SACMO) were provided treatment in primary care settings as an assistant to qualified doctors. On the other hand, medical technicians (MTs) and sanitary inspector had a very limited role in direct patient management [4].

In this study, the prevalence of HTN was higher than the DM among those screened and this finding aligned with the national findings of Bangladesh [21]. However, this nationally representative study did not include HPs as a study population. Another study among trainee doctors also revealed the same trend of HTN and DM [8]. In our study, the proportion of HTN screening was higher than the DM, thus more HPs were diagnosed as hypertensive than diabetics. In the current study, most of the HPs were treated for HTN and DM which is supported by a previous study Bangladesh among HPs [8]. Interestingly, the control rate of HTN among HPs (63.4%) of the current study was similar to a study conducted in the HPs (64.7%) of Cameroon [22]. Compared to HTN, half (31%) of the HPs controlled their DM which was much lower (53.7%) than the HPs of Iran [23]. We found a high prevalence of key risk factors among those diagnosed with HTN and DM namely physical inactivity, unhealthy diets (low fruit and/or vegetable intake and high salt intake), and overweight/obesity. The possibility of such high prevalence includes a lack of self-motivation and negligence. Regarding self-motivation, around 26% of doctors with medical problems reported feeling inhibited to seek treatment [24]. In this case, possibly the HPs neglected these risk factors or hold an idealistic view of their role in treating and fighting diseases [25]. Their idealistic view is that they are a 'caregiver', not 'caretaker'. Hence, they rarely seek treatment from other health professionals [24]. A previous study also reported that HPs do not practice what they preach and may not think that self-care is a priority [26]. Possibly, behavioral risk factors (physical inactivity and unhealthy diet) and some barriers to adhering to a healthy lifestyle synergistically contribute to overweight/obesity in HPs [27]. We found that avoidance of smoking is the only risk factor significantly associated with health professional categories. However, it was possibly elucidated due to the influence of gender (SSNs = 87.8% women, 12.2% men; PHPs = 85.3% men, 14.7% women). This gender-based association with smoking has already been reported by a study among HPs in Bangladesh [8]. In Bangladesh, culturally and religiously, women are not generally smokers and working in the health sector also reduces the possibility of being a smoker.

One of the major strengths of this study is its population of interest, who are very selective and rarely reported in global research, especially in low- and lower-middle-income countries. Another strength was the study design that used a 100% sample to collect information and therefore the estimates were not subject to sampling error. Here, HPs were recruited from all Upazila Health Complexes in the country covering 8 divisions and 64 districts. In Bangladesh, for the first time, healthy bahaviours, treatment, and control status about HTN and DM were assessed among health professionals that will guide the formulation of policy for health professionals to ensure their efficiency in the health system.

Despite these strengths, the current study has some limitations too. Here, self-reported risk factors may increase the risk of recalled bias, and the inclusion of HPs other than qualified doctors

reduce the generalizability of the findings. The diagnosis of DM might have overestimated or underestimated the real prevalence as we did not use the gold standard oral glucose tolerance test. Besides, few outliers were identified using univariate and multivariate outlier detection methods. However, we did not remove them as they did not influence our findings (S1 Data).

In terms of screening and treatment for HTN and DM, the HPs of Bangladesh showed satisfactory results. However, control of DM was half of the HTN that demands a comprehensive risk reduction policy including the component of self-motivation. This is because HTN and DM follow each other over time and complications may arise from any side. Again, very poor adherence to recommend lifestyle modification was observed among the HPs for the risk factors that warrant further study to find out the background barriers to such non-adherence. Therefore, based on the current findings, future interventions should emphasize smoking, physical activity and the dietary domain to sustain HPs productivity in their health system. For this, attitudes towards chronic diseases and their health should be changed as they are expected to be caregivers and not "caretakers". Hence, policymakers should develop separate screening and health education intervention packages including mental health for HPs to motivate them and allow them to lead an active and productive life.

## Supporting information

**S1 Data. Multivariate and univariate outlier detection.**
(PDF)

## Acknowledgments

The authors would like to acknowledge the Noncommunicable Disease Control Program, Directorate General of Health Services, Ministry of Health & Family Welfare, Government of the People's Republic of Bangladesh, for their kind cooperation to conduct the study.

## Author Contributions

**Conceptualization:** Lingkan Barua, Palash Chandra Banik, Mithila Faruque.

**Data curation:** Lingkan Barua, Palash Chandra Banik.

**Formal analysis:** Lingkan Barua, Palash Chandra Banik, Mithila Faruque.

**Investigation:** Mithila Faruque.

**Methodology:** Lingkan Barua, Palash Chandra Banik, Mithila Faruque.

**Project administration:** Palash Chandra Banik, Mithila Faruque.

**Resources:** Mithila Faruque.

**Supervision:** Mithila Faruque.

**Writing – original draft:** Lingkan Barua.

**Writing – review & editing:** Lingkan Barua, Palash Chandra Banik, Mithila Faruque.

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
