## [Decision Letter · Decision Letter 0]

5 Apr 2023

PGPH-D-22-01982

Healthy lifestyle practice, treatment, and control status of diagnosed hypertension and diabetes among the government nurses and para-health professionals of Bangladesh: A cross-sectional study

Dear Dr. Barua,

Thank you for submitting your manuscript to PLOS Global Public Health. After careful consideration, we feel that it has merit but does not fully meet PLOS Global Public Health’s publication criteria as it currently stands. Therefore, we invite you to submit a revised version of the manuscript that addresses the points raised during the review process.

Please clarify the following

1)the questions used to collect data on healthy behaviours (smoking etc) to control hypertension and diabetes. Did the questions explicitly ask if these behaviours were undertaken to control hypertension and diabetes

2)the questions used to collect data on "days since the last measurement of HBP and BG". Were these open ended questions? If categorical what where the categories used?

The manuscript will benefit from an extensive editing and proof reading process

We look forward to receiving your revised manuscript.

Kind regards,

Sherly Parackal, PhD

Academic Editor

Journal Requirements:

1. Please update your online Competing Interests statement. If you have no competing interests to declare, please state: “The authors have declared that no competing interests exist.”

2. Please provide separate figure files in .tif or .eps format only and ensure that all files are under our size limit of 10MB.

3. Please ensure that you refer to Tables 1 and 2 in your text as, if accepted, production will need this reference to link the reader to the table.

Reviewers' comments:

Reviewer's Responses to Questions

**Comments to the Author**

1. Does this manuscript meet PLOS Global Public Health’s publication criteria? Is the manuscript technically sound, and do the data support the conclusions? The manuscript must describe methodologically and ethically rigorous research with conclusions that are appropriately drawn based on the data presented.

Reviewer #1: Yes

Reviewer #2: Yes

2. Has the statistical analysis been performed appropriately and rigorously?

Reviewer #1: Yes

Reviewer #2: Yes

3. Have the authors made all data underlying the findings in their manuscript fully available (please refer to the Data Availability Statement at the start of the manuscript PDF file)?

Reviewer #1: Yes

Reviewer #2: Yes

4. Is the manuscript presented in an intelligible fashion and written in standard English?

Reviewer #1: Yes

Reviewer #2: Yes

5. Review Comments to the Author

Reviewer #1: This study focused on senior staff nurses and para-health professionals’ healthy lifestyle practice, treatment, and control status of hypertension and diabetes mellitus. This manuscript may provide meaningful insights to readers and merit publication after revision.

I have several comments.

# Introduction

1. Page 3, Line68: In this study, the authors assessed only hypertension and diabetes. Why did the author focus on these two diseases rather than other lifestyle-related diseases or mental health which has been often reported among health professionals? An explanation regarding this part of the study would support the significance of this study.

# METHODS

2. Page 3, Line 75: Please explain SSNs. Are SSNs certified qualifications?

# DISCUSSION

3. Page 8, Line 185: Author discussed that “t seems that HPs neglected these risk factors or hold idealistic view of their role in treating and fighting diseases.” What does the meaning of hold idealistic view of their role in treating and fighting diseases? Please explain to make the readers understand easily.

4. There were no differences between SSNs and PHPs, except for smoking. What do you consider these results? Were these results as you hypothesized? Please discuss about that.

5. Based on the results, how should health professionals be educated to maintain their health?_

Reviewer #2: This paper analyzed the data of senior staff nurses (SSNs) and para-health professionals (PHPs) to assess their healthy lifestyle practice, treatment, and control status of hypertension (HTN) and diabetes mellitus (DM). A sample of 1942 health professionals was collected from Dhaka's capital city. The prevalence of self-screening, diagnosis, treatment and control status

did not reveal any notable differences between SSNs and PHPs. Overall the author(s) find that control of HTN, DM and their risk factors is not satisfactory among the SSNs and PHPs of Bangladesh.

This is an interesting study with overall quite valuable findings. However, several issues in the manuscript require clarification or amendment.

1. Abstract: The abstract of the paper is too long. At the moment, it seems to be a part of introducing. The abstract should be concise and summarize the entire paper.

2. Introduction: The author(s) need to better motivate the study by adding what is already known about this topic in Bangladesh. Further, it would be appropriate to mention the key contributions of this study and how they differ from existing literature in Bangladesh.

3. Material and Methods:

a. The author(s) should use either "a survey or questioner" in the data collection part.

b. What is the motivation for using the sample only from the capital city, Dhaka?

c. The author(s) discussed in the abstract that the sample is from the capital city, Dhaka. However, the author(s) did not discuss it in the paper. Why is it so?

d. There is no information about the information sheet. Is it provided before the survey?

e. Please provide details on the "Declaration of Helsinki (October 2013)."

f. Can you please explain "residual confounders"? Unfortunately, it is not a common term for healthcare research.

4. Results:

a) The result section is more of a statistical exercise, where the author(s) differentiate based on different statistics. Therefore, the author(s) should explain it more simply so the reader can easily understand it.

b) The paper has two tables (Table 1 and Table 2). However, none of these is referred to anywhere in the paper.

c) Results need to be discussed in more detail and explained in easy words so the audience can understand them better.

5. Discussion: Most references are just cited in the discussion section. The author(s) should adequately link the cited papers in the discussion. The author(s) should state whether their results align with the prior literature or differ from theirs. It is good to write the discussion by looking at the similarities and differences between the literature and the strength and limitations.

6. Author should also add the limitations of the study.

6. PLOS authors have the option to publish the peer review history of their article (what does this mean?). If published, this will include your full peer review and any attached files.

**Do you want your identity to be public for this peer review?** For information about this choice, including consent withdrawal, please see our Privacy Policy.

Reviewer #1: No

Reviewer #2: No

---

## [Decision Letter · Decision Letter 1]

15 May 2023

PGPH-D-22-01982R1

Healthy lifestyle practice, treatment, and control status of diagnosed hypertension and diabetes among the government nurses and para-health professionals of Bangladesh: A cross-sectional study

Dear Dr. Barua,

Thank you for submitting your manuscript to PLOS Global Public Health. After careful consideration, we feel that it has merit but does not fully meet PLOS Global Public Health’s publication criteria as it currently stands. Therefore, we invite you to submit a revised version of the manuscript that addresses the points raised during the review process.

We look forward to receiving your revised manuscript.

Kind regards,

Sherly Parackal, PhD

Academic Editor

Journal Requirements:

Additional Editor Comments (if provided):

Overall: Please improve the written language of your manuscript and check for grammatical errors and sentence structures. For e.g. Chronic illness among health professionals is merely reported. Using the word rarely would better convey what you are trying to communicate. There are several other sentences that require revising for achieving good written standard.

Abstract: Please avoid repetitions for e.g. 100% sample. This should be referred to as a census rather than a 100% sample.

Methods: Please include clearly the methods adopted for measuring HTN and diabetes and anthropometry. The references and standards are good but readers need to know how this was done. How many measurements/readings were taken. Were any bias in measurements introduced by different people taking measurements. Who took the measurements, where they trained?

Objectives and results: your objective -risk factors modification for diagnosed cases is incorrect as you did not ask this question. Your explanation ...No, we did not explicitly ask if they tried to modify the behaviors to control hypertension and

diabetes. We thought if we asked explicitly, then the findings could be biased as they knew about the control measures as a health professional and, risk factors-specific questions leaded them to a common positive answer. We assumed as a health professional, their knowledge, attitude and practice about the risk factors and chronic diseases are different than the general population.... is not acceptable. Please change the wording of this aim, in the tables, results and discussion. What would be more appropriate is "assessing healthy behaviours among those diagnosed with diabetes and HTN" . Could the lack of key healthy behaviours among those diagnosed be a reason for poor control of HTN and diabetes? Since you did not ask the question whether they practiced these healthy behaviours for controlling diabetes or HTN, you need to rephrase your sentences accordingly. For e.g. Results: Line 141: Regarding healthy lifestyle practice to control HTN and DM... should be rephrased. As per Table 1: only 63.4% had a controlled HTN and Table 2: 31% had controlled diabetes. Your analysis includes all those who were diagnosed with HTN and diabetes, hence you cannot say that these healthy behaviors were practiced for controlling HTN or diabetes.

You need to expand on the limitations of the study. Apart from recall bias due to the self reported nature of the questionnaire data are there any other limitations? for e.g. in measuring HTN and diabetes or anthropometry? What about missing data and outliers? For e.g. your METS per week data has substantial outliers. How did you deal with this? You have reported this data as median, why?

Reviewers' comments:

Reviewer's Responses to Questions

**Comments to the Author**

1. If the authors have adequately addressed your comments raised in a previous round of review and you feel that this manuscript is now acceptable for publication, you may indicate that here to bypass the “Comments to the Author” section, enter your conflict of interest statement in the “Confidential to Editor” section, and submit your "Accept" recommendation.

Reviewer #1: All comments have been addressed

Reviewer #2: All comments have been addressed

2. Does this manuscript meet PLOS Global Public Health’s publication criteria? Is the manuscript technically sound, and do the data support the conclusions? The manuscript must describe methodologically and ethically rigorous research with conclusions that are appropriately drawn based on the data presented.

Reviewer #1: Yes

Reviewer #2: Yes

3. Has the statistical analysis been performed appropriately and rigorously?

Reviewer #1: Yes

Reviewer #2: Yes

4. Have the authors made all data underlying the findings in their manuscript fully available (please refer to the Data Availability Statement at the start of the manuscript PDF file)?

Reviewer #1: No

Reviewer #2: Yes

5. Is the manuscript presented in an intelligible fashion and written in standard English?

Reviewer #1: Yes

Reviewer #2: Yes

6. Review Comments to the Author

Reviewer #1: Thank you very much for your resubmitting the paper.

The reviewer asks the authors to discuss how they believe that the lack of differences between SSNs and PHPs was not due to differences in gender, but rather the lack of differences in job type.

The authors answered that SSNs is a government authorized position offered based on seniority, experience and also qualification (Diploma/BSc).

Do SSNs and PHPs have the same level of health care knowledge and skills?

Readers unfamiliar with the health care system and the education system in Bangladesh may be confused by this discussion.

For this reason, I pointed out that the authors should explain what SSNs is and how they differ from PHPs in the first peer review.

I am sorry, but since there is no discussion on this, I cannot support the education for the health care profession that the author states in his current conclusion.

We believe it is important in your paper to explain and discuss the situations in Bangladesh for the benefit of readers who are unfamiliar with it.

Reviewer #2: I am satisfied with the changes authors made in the revised draft.

7. PLOS authors have the option to publish the peer review history of their article (what does this mean?). If published, this will include your full peer review and any attached files.

**Do you want your identity to be public for this peer review?** For information about this choice, including consent withdrawal, please see our Privacy Policy.

Reviewer #1: No

Reviewer #2: **Yes: **Sumera Saeeed Akhtar

---

## [Decision Letter · Decision Letter 2]

12 Jul 2023

Healthy behaviours, treatment, and control status of diagnosed hypertension and diabetes among the government nurses and para-health professionals of Bangladesh: A cross-sectional study

PGPH-D-22-01982R2

Dear Dr. Barua,

We are pleased to inform you that your manuscript 'Healthy behaviours, treatment, and control status of diagnosed hypertension and diabetes among the government nurses and para-health professionals of Bangladesh: A cross-sectional study' has been provisionally accepted for publication in PLOS Global Public Health.

Best regards,

Loai Albarqouni, M.D. M.Sc. Ph.D.

Academic Editor

Reviewer Comments (if any, and for reference):

Reviewer's Responses to Questions

**Comments to the Author**

1. If the authors have adequately addressed your comments raised in a previous round of review and you feel that this manuscript is now acceptable for publication, you may indicate that here to bypass the “Comments to the Author” section, enter your conflict of interest statement in the “Confidential to Editor” section, and submit your "Accept" recommendation.

Reviewer #1: All comments have been addressed

Reviewer #2: All comments have been addressed

2. Does this manuscript meet PLOS Global Public Health’s publication criteria? Is the manuscript technically sound, and do the data support the conclusions? The manuscript must describe methodologically and ethically rigorous research with conclusions that are appropriately drawn based on the data presented.

Reviewer #1: Yes

Reviewer #2: Yes

3. Has the statistical analysis been performed appropriately and rigorously?

Reviewer #1: Yes

Reviewer #2: Yes

4. Have the authors made all data underlying the findings in their manuscript fully available (please refer to the Data Availability Statement at the start of the manuscript PDF file)?

Reviewer #1: Yes

Reviewer #2: Yes

5. Is the manuscript presented in an intelligible fashion and written in standard English?

Reviewer #1: Yes

Reviewer #2: Yes

6. Review Comments to the Author

Reviewer #1: I am satisfied with the changes authors made in the revised draft.

Reviewer #2: I am satisfied with the quality of the current version of the article.

7. PLOS authors have the option to publish the peer review history of their article (what does this mean?). If published, this will include your full peer review and any attached files.

**Do you want your identity to be public for this peer review?** For information about this choice, including consent withdrawal, please see our Privacy Policy.

Reviewer #1: No

Reviewer #2: **Yes: **Sumera Saeed Akhtar
